# Trauma and Remembering: From Neuronal Circuits to Molecules

**DOI:** 10.3390/life12111707

**Published:** 2022-10-26

**Authors:** Szabolcs Kéri

**Affiliations:** 1Department of Cognitive Science, Budapest University of Technology and Economics, 1111 Budapest, Hungary; keri.szabolcs@ttk.bme.hu; Tel.: +36-1463-1273; 2National Institute of Mental Health, Neurology, and Neurosurgery, 1145 Budapest, Hungary; 3Department of Physiology, Albert Szent-Györgyi Medical School, University of Szeged, 6720 Szeged, Hungary

**Keywords:** trauma, memory, learning, posttraumatic stress disorder (PTSD), amygdala, engram, MDMA, psychedelics

## Abstract

Individuals with posttraumatic stress disorder (PTSD) experience intrusions of vivid traumatic memories, heightened arousal, and display avoidance behavior. Disorders in identity, emotion regulation, and interpersonal relationships are also common. The cornerstone of PTSD is altered learning, memory, and remembering, regulated by a complex neuronal and molecular network. We propose that the essential feature of successful treatment is the modification of engrams in their unstable state during retrieval. During psychedelic psychotherapy, engrams may show a pronounced instability, which enhances modification. In this narrative review, we outline the clinical characteristics of PTSD, its multifaceted neuroanatomy, and the molecular pathways that regulate memory destabilization and reconsolidation. We propose that psychedelics, acting by serotonin-glutamate interactions, destabilize trauma-related engrams and open the door to change them during psychotherapy.

## 1. Introduction

Posttraumatic stress disorder (PTSD) is one of the most researched and controversial topics in contemporary psychiatry. In a significant proportion of people with repeated and severe trauma, transient or subclinical PTSD-like symptoms appear, but the complete syndrome is diagnosed in a relatively small proportion of the affected population [1]. It is noteworthy that more than half of PTSD patients also have severe comorbidities, such as addictions or major somatic illnesses [2,3]. Given the limited effectiveness of therapeutic options, understanding the pathophysiological underpinnings of PTSD is essential.

In this narrative review, we highlight memory alterations associated with PTSD. The core concept of the paper is that PTSD is a disorder of learning, memory, and remembering. We will then examine the neural basis and molecular mechanisms that regulate engram (dynamic memory trace) destabilization and consolidation. We argue that the key effect of psychedelic psychotherapy, which has received increasing attention in the treatment of PTSD, is to transform engrams and reorganize autobiographical memory.

## 2. The History and Diagnosis of PTSD

The roots of PTSD date back to the “father of history,” Herodotus, who described the case of Epizelus, a soldier who had dissociative blindness due to combat trauma in the Battle of Marathon (490 BCE) [4]. During the middle age, the ethos and morals of knighthood and self-sacrifice were deep sources of trauma and loss (*Livre de chevalerie*, 1315), which became more prevalent in the violent wars of the 18th–20th century in Europe and America. During the seven-year war (1756–1763), the Austrian physician Josef Leopold Auenbrugger reported a severe mental condition characterized by fear, terror, anxiety, and a melancholic strive of nostalgia and homesickness. It was similar to Da Costa’s soldiers’ heart, which was based on the detailed medical history of 300 members of the military personnel involved in the American Civil War (1861–1865). In addition to the mental problems, these people complained of chest pain, fatigue, and shortness of breath. In the First World War (1914–1918), approximately 10 percent of soldiers presented the symptoms of shell shock (anxiety, dizziness, tremor, and enhanced sensitivity to sensory stimuli), a similar rate to battle fatigue and battle neurosis in the Second World War (1939–1945) [4,5,6].

From a psychiatric point of view, the critical event was the Vietnam War (1955–1975), which eventually led to the birth of the nosological category of PTSD in the 3rd edition of the Diagnostic and Statistical Manual of Mental Disorders (DSM-III) in 1980 [7]. After decades of dispute on the reliability and validity of PTSD, the DSM-5 did not provide relief for the clinician [2]. With the introduction of the new dimension of “negative alterations in mood and cognition” and the dissociative subtype of PTSD, we have 636,120 symptom combinations to establish the diagnosis [8]. Moreover, the 11th edition of the International Classification of Diseases (ICD-11), but not the DSM-5, introduced the new category of complex PTSD characterized by disorders in identity, interpersonal relationships, and affective regulation [9]. Therefore, it is not surprising that the diagnostic systems have a moderate degree of agreement [2].

The prevalence of PTSD (World Mental Health Survey cross-national lifetime prevalence in the total population: 3.9%; among trauma-exposed individuals: 5.6%) depends on trauma type (e.g., higher risk for cumulative exposure, rape, physical assault), age, sex, socioeconomic status, and pre-trauma health status [1]. Given that traumatic events are quite common (70–90% in a given population) relative to the prevalence of PTSD, the majority of individuals in a community possess various coping mechanisms and resilience to deal with the psychological and physical consequences of severe adverse life events [3,10,11,12].

The interactions among genetic vulnerability (PTSD heritability: 30–40%), socioeconomic status, social support, trauma, and physiological changes in PTSD (e.g., low-grade peripheral inflammation) are complex and multifaceted [11,13]. However, it seems that polygenic components correlate with low socioeconomic status, a common mediator for the likelihood of trauma, lack of social support, physiological changes, and the development of PTSD [13].

## 3. Clinical Manifestation

According to DSM-5, individuals with PTSD directly experience or witness actual or threatened death, serious injury, or sexual violence [14]. Later, they reexperience the traumatic event in the form of intrusive, sensorial, and emotional memories, flashbacks, and nightmares filled with fear, horror, and intensive autonomic reactions (e.g., palpitation, sweating, and shortness of breath). These intrusive experiences result in multiple forms of active and passive avoidance, including thought and memory suppression and avoiding places and activities that may remind the patient of the traumatic event. The third classic symptom dimension of PTSD is heightened arousal (hypervigilance, sleep disorder, and enhanced startle responses) [14].

The DSM-5 separately underlines the importance of negative cognitions and mood, which comprises general signs and symptoms of weak memory formation, negative beliefs and expectations, cognitive distortions leading to abnormal blaming and feelings of threat, and negative emotionality (fear, anger, shame, guilt, diminished interest, and lack of positive feelings) [14,15]. In addition, the DSM-5 offers further specifications regarding dissociative symptoms. The patient may feel depersonalization (feeling detached from one’s mental processes) and derealization (feeling of unreality of surroundings) [14].

In complex PTSD, an exclusive ICD-11 diagnosis, individuals survive frequent early, persistent, and severe trauma. In addition to the classic PTSD phenomena, additional symptoms of disordered self-concept (e.g., poor self-esteem, acceptance, and commitment), impaired affective regulation (emotional numbing or over-reactivity), and hassle with interpersonal relationships are also present in complex PTSD. Recent data suggest that 1–8% of the population has complex PTSD, and in mental health facilities, even 50% of the patients can have this diagnosis [9].

## 4. Learning and Memory in PTSD: From Experiencing to Neuronal Circuits

From our perspective, it is critical how patients with PTSD learn and remember. A vast amount of research shows that individuals with PTSD are susceptible to fear learning (aversive associative conditioning), overgeneralization of fear memories to neutral contexts (e.g., expecting an explosion of a bomb in a peaceful village), and lessened extinction (diminishing of aversive memories during repetitions over time without negative consequences) [16,17,18,19,20].

By definition, PTSD patients experience vivid, emotional, and intrusive memories of the trauma [14]. However, they often report poor attention, inability to remember specific details of events (dissociative amnesia), and inefficient learning related to latent avoidance [21,22,23]. Fighting with intrusive memories, rumination, internal avoidance, and impaired attentional control leads to reduced autobiographical memory specificity: the discrete time, location, and distinctive emotional/social characteristics of internal representations of places, people, and events are diminished [17,24]. Multiple mechanisms may contribute to decreased autobiographic memory specificity, including rumination, functional avoidance, and executive dysfunctions. The greying out of autobiographical memory has a definitive impact on social functioning, planning, problem-solving (prospective memory), emotion regulation, and quality of life [24].

Interestingly, there is a definitive overlap between the neuronal representation of autobiographical memory specificity and PTSD [24]. The hippocampal formation and its main gateway to the cortex (cuneus, precuneus, posterior cingulum) regulate the self-relevant physical and social details of events in an appropriate space–time context by delineating distinct engrams (pattern separation). The medial prefrontal cortex is crucial for self-referential processing and emotional salience regulation. Finally, the dorsal prefrontal cortex serves executive functions, memory retrieval, and engram reconstruction [24] (Figure 1).

## 5. Large-Scale Neuronal Networks in PTSD

It is essential to underline that a widespread neuronal network, well beyond the classic fear learning circuit, is responsible for altered learning and memory in PTSD [19,25,26,27,28] (Figure 1). Sensory input coding the unconditioned and conditioned stimuli (i.e., the traumatic event and the associated environmental cues) reaches the lateral and basal parts of the amygdala before cortical processing [29,30]. However, neuronal activity is relatively short in these areas, and input from the dorsal anterior cingulate cortex is necessary to maintain threat-related information processing. The dorsal anterior cingulate cortex also activates the striatum in the basal ganglia, contributing to threat-related behavioral actions [26,31]. On the other hand, the central nucleus of the amygdala, receiving information from the basolateral amygdala regions implicated in associative conditioning of aversive and neutral stimuli, sends fibers to the brainstem (e.g., the periaqueductal grey matter) and hypothalamic centers, eliciting trauma- and stress-related physiological responses [32].

How can the brain regulate the cingulate–amygdala fear system? Three critical networks maintain the balance and counter-regulate stress-related responses. First, the ventromedial prefrontal cortex inhibits the amygdala and the dorsal anterior cingulate cortex [16,32,33,34,35]. This is essential for extinction, which is not a mere forgetting because it requires the emergence of new engrams of safety memory. The safety memory then inhibits engrams of threat memory [32].

Second, the hippocampal formation is essential for establishing and maintaining an appropriate context of time and location for memories [32,36] (Figure 1). For example, the traumatic event that happened several months ago in a distant town is bound to this context, and hence could not be experienced as happening here and now. The failure of hippocampal coding of the spatiotemporal context is a possible mechanism of the intrusive reexperiencing of traumatic episodes in PTSD, together with a failure to effectively encode new events [36].

General models of hippocampus-dependent associative learning focus on cue–context links. In these models, the primary role of hippocampal associative learning is bridging foreground cues and background context to obtain a nuanced representation of an event. Many facets of memory alterations in PTSD can be attributed to the impaired integration of cue (e.g., injured people) and context (e.g., a city landscape), resulting in inappropriate representations of what, where, and when something happened [37,38].

The hippocampal formation is a hub in large-scale neuronal networks. It is responsible for self-referential processing and salience attribution via the recruitment of autobiographical memories and their emotional content, comprising a complex architecture of cues and contexts [39,40]. In addition to decreased hippocampal volume, patients with PTSD often show poor performance and weakened hippocampal activity during fear renewal and extinction recall tasks [41]. Moreover, the functional integration of the hippocampal formation into default mode and salience networks is also disrupted, contributing to higher-level social dysfunctions in PTSD [41].

The beliefs and expectations of the patients are also linked to hippocampal functioning [42]. A combination of computational modeling and functional neuroimaging suggests that patients with PTSD rely on their beliefs to control hippocampal activity during memory suppression [43]. Furthermore, error signals between the expectations/beliefs and actual events (prediction errors) were associated with the emergence of unwanted intrusions and avoidance behavior [43,44].

The third central system implicated in PTSD is the dorsolateral prefrontal executive network that mediates cognitive control to redirect attention from threat-related cues and thoughts to other positive and salient events [26,39,40] (Figure 1). Neurochemical and synaptic alterations are the opposite in prefrontal and limbic areas: in the prefrontal cortex, glutamatergic changes lead to decreased synaptic connectivity, whereas in the amygdala, monoaminergic mechanisms induce synaptic hyperplasticity and hyperconnectivity [45]. From a large-scale neural network perspective, decreased activity and connectivity were detected in the central executive network (attentional regulation, cognitive deficits) and default mode network (dissociation, avoidance, and intrusive thoughts). In contrast, there is increased activity and connectivity in the salience network (heightened threat detection and impaired regulation of the central executive and default mode network) [39,40,45].

Recent evidence suggests an intriguing epigenetic alteration in the prefrontal-amygdala circuit, resulting in over-consolidating fear memories in PTSD [46]. Specifically, the downregulation of a histone methyltransferase in the prefrontal cortex promoted fear expression by enhancing memory consolidation. Genes implicated in synaptogenesis showed increased expression in the prefrontal-basolateral amygdala circuit following the epigenetic changes, which may be a critical cellular factor in altered learning and memory [46].

Genome-wide association studies also suggested altered gene expression in the anterior cingulate-prefrontal system, behaviorally linked to a general mood-anxiety-neuroticism factor in PTSD [47]. Genetically regulated transcriptomic changes indicated two genes that consistently showed altered expression in the prefrontal, cingulate, cortical, and limbic regions: *DND1P1* and *ARL17A*. The *DND1P1* gene encodes a protein binding to microRNA-targeting sequences of mRNAs, and inhibits the microRNA-mediated repression of translation. This mechanism may be implicated in the regulation of genes participating in synaptic plasticity. *ARL17A* (ADP Ribosylation Factor Like GTPase 17A) encodes a GTP binding protein that regulates the functioning of several neurotransmitter receptors and cellular trafficking [47].

In summary, traumatic remembering includes multiple factors and mechanisms: enhanced associative learning of fear-related cues, impaired encoding of spatiotemporal context, over-generalization and enhanced consolidation of fear memories, and weak extinction. In addition to the traumatic event, genetic and epigenetic changes contribute to the abnormal formation of fear memories.

## 6. Reconsolidation of Fear Memories: A Potential Mechanism of Action for Psychedelic Substances in PTSD

Following a large body of anecdotal reports on the use of psychedelic-associated psychotherapy in the treatment of PTSD [48,49], Mitchell et al., (2021) demonstrated the effectiveness of this treatment in a randomized, double-blind, placebo-controlled, phase 3 trial [50]. They investigated 90 patients with severe PTSD to explore the efficacy and safety of 3,4-methylenedioxymethamphetamine (MDMA, ecstasy)-assisted psychotherapy. They found significantly decreased PTSD symptoms in the MDMA group relative to the placebo, with a vast, clinically unusual effect size (*d* = 0.91) [50]. Mitchell et al. (2021) concluded that MDMA-assisted therapy is a potential breakthrough treatment for severe PTSD with multiple comorbidities. However, the mechanism of the robust therapeutic effect of psychedelics is not precisely known [51]. One possible solution lies in the reconsolidation of retrieved engrams, which are dynamic assemblies of neuronal networks serving memory traces [20,32,52].

In fear learning, the new engram consisting of a fear-provoking unconditioned stimulus (e.g., an explosion) and the conditioned context (e.g., the place where it happened and the people around the explosion) form a new active engram that consolidates into an allocated inactive engram via amygdala-hippocampal-cortical interactions [19,29,34,53]. During retrieval, the engram is destabilized, and there is a chance to modify it, for example, by extinction or reconstructing the content [54,55]. The fundamental principle of the reconsolidation hypothesis is that, during remembering, engrams turn into a destabilized state. Therefore, via psychological and pharmacological modulation, the engram can be changed. Then, this modified content will reconsolidate into an altered engram serving an adaptive behavior instead of fear, intrusive reexperiencing, and avoidance [32]. In popular terms, journalists often write about “erasing” and “creating” memories.

In an animal model, Hake et al. (2019) demonstrated that MDMA administered specifically during the reconsolidation phase reduced conditioned fear [56]. However, MDMA did not affect extinction. The authors concluded that MDMA augments psychotherapy by modifying the reconsolidation of fear memories in PTSD [56,57]. Interestingly, the same effect was found for another psychedelic, N,N-dimethyltryptamine (DMT, Ayahuasca) [58] but not for psilocybin [59].

MDMA and other psychedelic drugs have multiple mechanisms that may counteract PTSD-related pathophysiological changes and profoundly impact memory reconsolidation [60]. MDMA, psilocybin, and ketamine possess immunosuppressive and anti-inflammatory effects by reducing cytokine secretion and immune cell activation, which may impact memory formation, reconsolidation, and the specificity of autobiographical memories [61,62,63,64]. Enhanced peripheral inflammation and altered immune responses are cardinal features of PTSD as a general evolutionary response to threat and danger [62,65,66]. Data from animal studies suggest that the administration of Tumor Necrosis Factor-α (TNFα), a first-line cytokine secreted by macrophages, into the dorsal hippocampus disrupted the retrieval of contextual fear memory, decreased freezing responses, and impaired the retrieval and reconsolidation of spatial memory [67]. In addition, hippocampal TNFα applied before retrieval ceased *c-fos* early intermediate gene expression. Therefore, TNFα inhibits the reconsolidation of engrams in the hippocampal formation [67].

Second, MDMA induces rapid secretion of cortisol and other hormones (e.g., oxytocin) and may facilitate the downregulation of hypersensitive cortisol receptors [68]. Inhibiting cortisol synthesis during early morning sleep improves reactivated memories, which indicates that enhanced glucocorticoid signals disrupt reconsolidation [69]. It is widely believed that hypersensitivity of the hypothalamic-pituitary-adrenal gland (HPA) axis is a crucial feature of PTSD [70]. The normalization of cortisol receptor hyperactivity may also contribute to changes in memory reconsolidation, improved context processing, and volume changes in hippocampal formation. Astil Wright et al., (2021) concluded that hydrocortisone, Reconsolidation of Traumatic Memories therapy, and cognitive task interference during memory reactivation of intrusive contents were effective in treating PTSD [52]. This suggests that unstable engrams can similarly be modified by cognitive interventions (attentional distraction and sensory interference) and by stimulating glucocorticoid receptors.

## 7. Reconsolidation of Engrams and the Cellular Mechanism of Psychedelics

From a theoretical and clinical point of view, it is indispensable to understand how psychedelics and other similar substances modify the reconsolidation of engrams at the intracellular level [53,71]. MDMA is a potent monoamine reuptake inhibitor. In addition to blocking the serotonin, norepinephrine, and dopamine transporter in the presynaptic terminal, MDMA also inhibits type 2 vesicular monoamine transporter (VMAT-2), resulting in a marked increase of monoamine concentration in the synaptic cleft. In the postsynaptic membrane, MDMA activates type 1A and 2A serotonin receptors (5-HT1A, 5-HT2A) [53,72,73]. 5-HT2A agonism is a common mechanism of serotonergic psychedelics, including psilocybin (“magic mushroom”, 4-phosphoryloxy-N, N-dimethyltryptamine) and Ayahuasca (N,N-dimethyltryptamine, DMT) [60]. Lysergic acid diethylamide (LSD) is only a partial agonist on 5-HT2A, promoting beta-arrestin activation and slow receptor desensitization, but a full agonist at 5-HT1A and several dopamine receptors (D1, D2, and D4) [53,74].

Although ketamine, an antagonist of N-methyl-D-aspartate (NMDA) glutamate receptors and an indirect stimulator of α-amino-3-hydroxy-5-methyl-4-isoxazolepropionic acid (AMPA) glutamate receptors, has a different mechanism of action, the sigma-receptor may form a bridge with serotonergic psychedelics [53,75]. Sigma receptors, previously considered opioid receptors, are abundant cell surface proteins with multiple actions on neuronal and peripheral tissue functions [76]. Sigma-1 receptors modulate NMDA-mediated synaptic plasticity via a calcium-dependent potassium channel when stimulated by the serotonin-related DMT [77,78]. DMT-sigma 1 activation blocks voltage-gated sodium currents in neurons, induces hypermobility in mice, and may contribute to pleiotropic effects, including neuronal protection, plasticity, and the modulation of inflammation and immunity. Surprisingly, MDMA also activates sigma-1 receptors with a behavioral effect similar to DMT in animal models [79]. In conclusion, serotonergic and dissociative (NMDA-antagonist) psychedelics act in cooperation at the cellular level [74]. These mechanisms are highly relevant in PTSD.

How are these molecular mechanisms related to memory modulation? There are two major intracellular pathways: The first is for memory reconsolidation, including the synthesis of new proteins and structural synaptic plasticity. The second route is memory destabilization during retrieval, involving proteasomes at which scaffolding protein degradation occurs [53]. The memory reconsolidation route is dominantly activated by the PI3K (Phosphoinositide 3-kinases)—mTOR (mammalian target of rapamycin)—p70S6K (ribosomal protein S6 kinase beta-1) intracellular pathway, together with the Wnt/beta-catenin system [80,81]. In addition, both metabotropic glutamate and 5-HT2A receptors activate PI3K [82].

Moreover, 5-HT2A receptors form dimers with type 2 metabotropic glutamate receptors (mGlu2), together with other G-protein coupled receptors [82] (Figure 2). The serotonin-glutamate receptor complex induces the phosphorylation of the mGlu2 receptor on a serine residue (Ser843) when 5-HT2A is stimulated by psychedelics [74]. This receptor crosstalk represents a direct interaction between the serotonin and glutamate systems to boost the synthesis of new proteins in the synapses and enhance memory reconsolidation.

P70S6K is one of the terminal factors in activating ribosomes, where new proteins are synthesized during synaptic plasticity [83]. Intriguingly, in individuals with PTSD, we found a profoundly decreased expression of the S6 kinase gene, which interacted with the effect of hyperactive cortisol receptors (FKBP5 regulation) on memory, hippocampal structure, and response to cognitive-behavioral therapy [84,85]. Furthermore, the cortisol receptor hypersensitivity tended to normalize during the treatment, whereas p70S6K expression did not exhibit significant changes [84,85].

The other main activating route of the memory reconsolidation molecular pathway is the cAMP-PKA (protein kinase A) and calcium-PKC (protein kinase C) system. Both converge on the MEK (mitogen-activated protein kinase kinase)—ERK kinase (extracellular signal-regulated kinases) cascade resulting in the phosphorylation and activation of the transcription factor CREB (cAMP response element-binding protein) [53,86]. The resulting expression of Zif268, a zinc-finger protein, is a cornerstone of synaptic protein synthesis, hippocampal long-term potentiation, and memory formation [87]. Moreover, Zif268 controls the maturation and assembly of hippocampal neurons into functional networks serving memory engrams [88]. Recently, it has been shown that hippocampal Zif268 is necessary for reconsolidating recognition memory [89]. Psychedelics activate the cAMP-PKA pathway via 5-HT1A receptors, whereas 5-HT2A receptors recruit the calcium-PKC in concert with the NMDA receptors [53].

The second, less known molecular cascade leads to scaffolding protein degradation and memory destabilization (Figure 2). The primary extracellular activator of this pathway is the slow decay GluN2B subunit of the NMDA receptor, acting separately from other receptor subtypes [53,90]. In this pathway, ubiquitin–proteosome degrading synaptic scaffolding proteins are mainly inflected by the calcium/calmodulin-dependent protein kinase II (CAMKII) [91]. It has been demonstrated that proteasome activity is elevated in the amygdala following the retrieval of contextual fear memory, suggesting synaptic protein degradation and engram destabilization. The inhibition of CAMKIII eliminated proteasome activation [91]. According to Milton et al. (2013), there is an intricate molecular balance between the destabilization and restabilization of engrams [92]. The NMDA subunit GluN2B regulates destabilization, whereas the GluN2A subunit performs restabilization [92]. CAMKII directly binds to the GluN2B subunit and regulates synaptic plasticity [93].

Early studies indicated that in rats, MDMA prevented the increased expression of the GluN1 subunit of the NMDA receptor during learning, together with a weak availability of CAMKII in the membrane [94]. Moreover, at the behavioral level, passive avoidance was diminished in the same animal model. There is now abundant evidence that psychedelics acting via 5-HT2A agonism impact CAMKII that also forms a bridge with the ERK—CREB system, and enhances the widespread expression of neuronal plasticity genes in the neocortex and hippocampus, the so-called “rapid psychoplastogenic changes” [95].

We propose that serotonin–glutamate receptor heterodimers, with a particular reference to 5-HT2A- mGlu2, have an intricate role in the mechanism of psychedelics’ action concerning memory modulation. These dimers operate via both Gq and Gi proteins. 5-HT—glutamate activations enhance CAMPKII via Gq proteins, whereas they inhibit cAMP—PKA via Gi proteins [74,82]. Both mechanisms contribute to the activation of scaffolding protein degradation at the proteasomes and the consequent destabilization of engrams [53,74,83].

## 8. Conclusions

The success of MDMA-assisted psychotherapy in treating PTSD could be explained by the profound destabilization of retrieved traumatic engrams, and the reconsolidation of newly modified engrams with a more positive emotional valence. This effect requires the coordinated activity of serotonergic and glutamatergic mechanisms, converging on a complex network of intracellular molecular pathways [53]. From an anatomical point of view, these processes impact the functional and microstructural reorganization of a large-scale neuronal network, including various amygdala nuclei, ventromedial and dorsolateral prefrontal cortex, anterior cingulate cortex, and the hippocampal formation. In addition, at the neurocognitive and affective level, attention to threat, associative aversive conditioning, implicit and intentional emotion regulation flexibility, and memory cue–context modulation may all be engaged [26].

The pharmacological adjustment of engram dynamics opens the door to receiving, processing, and incorporating new information during the psychotherapeutic session, enhancing bonding and the corrective emotional–interpersonal experience. Patients receiving psychedelic-associated therapy often report heightened openness, increased trust toward the therapist, less fear and internal avoidance, and a better ability to extinct, reframe and integrate traumatic memories [96].

Pharmacological modulation of the brain networks that receive information during psychotherapy is a substantial challenge for the therapeutic community. An adequately controlled and skilled psychological intervention can achieve a remarkable therapeutic effect, whereas inappropriate therapeutic processes increase the likelihood of iatrogenic effects and re-traumatization.

We must bear in mind that the application of psychedelics is not without danger on both the psychological and biological levels [97]. First, although MDMA and other similar substances are considered entactogens and empathogens, promoting positive social emotions, negative feelings may also be experienced, and the mental status of individuals with previous psychiatric history may decline [98]. Second, the robust effect on the endocrine system should be considered. Increased oxytocin levels might promote treatment effects by enhancing trust and cooperation, but increased cortisol and testosterone levels may contribute to the stress response and weakened impulse control [99]. Finally, following the administration of MDMA, during the rebound and recovery phase, users can experience depressive symptoms [97]. However, under controlled clinical conditions, the adverse effects can be minimized. The core conclusion is that: “This highlights the importance for clinicians and therapists to keep to the highest safety and ethical standards. It is imperative not to be overzealous and to ensure balanced media reporting to avoid future controversies, so that much needed research can continue” [100].

## Figures and Tables

**Figure 1 life-12-01707-f001:**
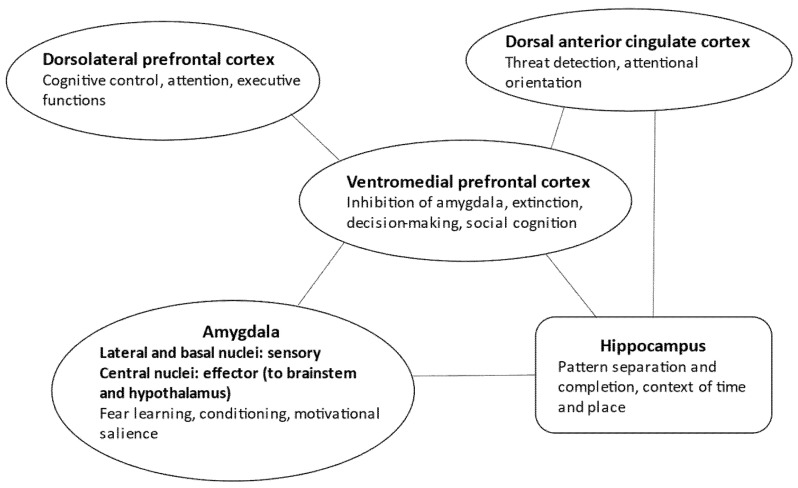
Neural circuits in PTSD.

**Figure 2 life-12-01707-f002:**
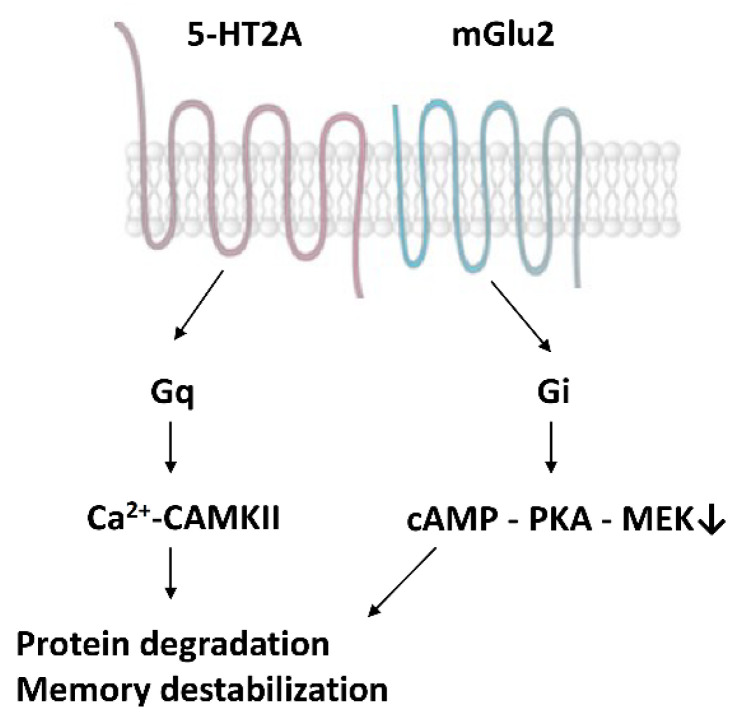
Heterodimers of serotonin (5-HT2A) and glutamate (mGlu2) receptors [53,74].

## Data Availability

Not applicable.

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
