# Peer review of "Trauma and Remembering: From Neuronal Circuits to Molecules"

_life, 2022, doi:10.3390/life12111707_

Round 1
Reviewer 1 Report
The manuscript is well written and the topic critically addressed.
I suggest only minor alterations:
Include Figure reference in the main text.
In line 325 add “with” in: “form dimers “with” type 2…”
Figure 2 need a resolution/quality improvement as well as reference to literature.
In line 358 change NR2B for GluN2B.
In line 369 change NR1 for GluN1.
In line 374 change “psychoplasticogenic” for “psychoplastogenic” as referred in [95]
In line 379 references need formatting.
Author Response
Thank you very much for the expert review of the manuscript. Below please find each point raised (Q) and the corresponding modifications in the text. The changes have been highlighted in red in the text.
Q1: „Include Figure reference in the main text.”
The Figures are now referenced in the text.
Q2: „In line 325 add “with” in: “form dimers “with” type 2…”
We added this word.
Q3: „Figure 2 need a resolution/quality improvement as well as reference to literature.”
The resolution has been improved and references are added not only in the text, but also in the figure legend.
Q4: “In line 358 change NR2B for GluN2B.”
The terminology of the receptor unit has been modified accordingly.
Q5: “In line 369 change NR1 for GluN1.”
The terminology of the receptor subunit has been changed accordingly.
Q6: “In line 374 change “psychoplasticogenic” for “psychoplastogenic” as referred in [95]”
The word has been corrected.
Q7: “In line 379 references need formatting.”
The references have been formatted according to MDPI style.

Reviewer 2 Report
The author summarized the current understanding of PTSD in this manuscript by discussing memory's important role in PTSD and went through the related neural circuit, mechanism, and possible treatment targets.
The manuscript is rich in content, well-written, and well-organized, with adequate references. I really enjoyed reading this manuscript.
I suggest this manuscript be accepted in its current form.
Author Response
Thank you very much for the review of the manuscript. I am happy that the material summarized in the paper is useful, and that you liked the paper. As highlighted, the text has been checked for language and typos.